
Rapid Wastage of the Hazen Plateau Ice Caps, Northeastern Ellesmere Island, Nunavut, Canada
Mark C. Serreze[1], Bruce Raup[2], Carsten Braun[3], Douglas R. Hardy[4] and Raymond S. Bradley[4]
[1]Department of Geography, and Cooperative Institute for Research in Environmental Sciences,
University of Colorado, Boulder Colorado, USA
[2]Cooperative Institute for Research in Environmental Sciences, University of Colorado, Boulder Colorado,
USA
[3]Geography and Regional Planning / Environmental Science, Westfield State University, Westfield,
Massachusetts, USA
[4]Department of Geosciences, University of Massachusetts, Amherst Massachusetts, USA

26                                            **Abstract**

Two pairs of small stagnant ice bodies on the Hazen Plateau of northeastern Ellesmere Island, the St.
Patrick Bay ice caps and the Murray and Simmons ice caps, are rapidly shrinking, and the remnants of
the St. Patrick Bay ice caps are likely to disappear entirely within the next five years.  Vertical aerial
photographs of these Little Ice Age relics taken during August of 1959 show that the larger of the St.
Patrick Bay ice caps had an area of 7.48 km$^2$, and the smaller one 2.93 km$^2$;  the Murray and Simmons
ice caps covered 4.37 km$^2$ and 7.45 km$^2$ respectively.  Outlines determined from ASTER satellite data for
July 2016 show that, compared to 1959, the larger and the smaller of the St. Patrick Bay ice caps had
both been reduced to only 5% of their former area, with the Murray and Simmons ice caps faring better
at 39% and 25%, likely reflecting their higher elevation.  Consistent with findings from other glaciological
studies in the Queen Elizabeth Islands, ASTER imagery in conjunction with past GPS surveys documents a
strikingly rapid wastage of the St. Patrick Bay ice caps over the last 15 years.  These two ice caps shrank
noticeably even between 2014 and 2015, apparently in direct response to the especially warm summer
of 2015 over northeastern Ellesmere Island.  The well-documented recession patterns of the Hazen
Plateau ice caps over the last 55+ years offer an opportunity to examine the processes of plant
recolonization of polar landscapes.
**Keywords:** Arctic, ice caps, mass balance, Little Ice Age, Hazen Plateau, Ellesmere Island, ASTER

**1. Introduction**

The Hazen Plateau of northeastern Ellesmere Island, Nunavut, Canada, is a rolling upland, with
elevations rising from about 300 meters above sea level near Lake Hazen to over 1000 m along the
northeast coast of the island. The plateau is unglaciated with the exception of two pairs of small
stagnant ice caps - the unofficially-named St. Patrick Bay ice caps, and, 110 km to the southwest, the
Murray and Simmons ice caps (**Figure 1**).  They are collectively referred to here as the Hazen Plateau ice
caps. As of 2001, the larger St. Patrick Bay ice cap ranged in elevation between about 880 m and 720 m
above sea level, with the smaller one spanning 820 m to 700 m.  The Murray and Simmons ice caps lie in
higher terrain; in 2001, both fell between about 1100 m and 1000 m above sea level.  The Hazen Plateau
ice caps are interpreted as forming and attaining their maximum extents during the Little Ice Age (LIA, c.
1600-1850) (Koerner, 1989).  Like much of the Queen Elizabeth Islands, the Hazen Plateau  is presently a
polar desert;  annual precipitation is typically only 150-200 mm, with a late summer and early autumn
maximum (Serreze and Barry, 2014).  Summer precipitation may be variously rain or snow. Summers are
very cool but variable; assessed as part of a multiyear glaciological study (Braun et al., 2004), the
average 10 m July air temperature at the Murray ice cap summit (1100 m) measured for the years 1999
through 2001, respectively, was 4.0$^o$C, 0.2$^o$C and 1.6$^o$C.
This paper documents the behavior of the Hazen Plateau ice caps over the past 55+ years in the context
of other glaciological studies in the Canadian Arctic.   The analysis is based on a combination of past
work using aerial photography, direct mass balance measurements from several field investigations, and
GPS surveys of ice cap areas collected as part of these investigations – along with new information on
ice cap areas using data at 15 m resolution from the ASTER (Advanced Spaceborne Thermal Emission
and Reflection Radiometer) instrument.  ASTER flies onboard the NASA's Earth Observing System Terra
satellite, launched in December 1999.  It provides reflectance at a 15 m resolution and is a key asset of
the international GLIMS initiative (Global Land Ice Measurements from Space) for mapping glacier
outlines (Raup et al., 2007; Kargel et al., 2014).

**2. Previous Work**

Table 1 lists all available direct mass balance estimates of the ice caps (in meters water equivalent, or
w.e.).  Table 2 provides all available estimates of ice cap areas ($km^2$).  The first information on the St.
Patrick Bay ice caps that we are aware of is oblique aerial photographs taken in late July of 1947 as part
of the U.S. Operation Polaris Trimetregon Survey.  These photographs show the ice caps standing out
prominently against the snow-free tundra surface. Vertical aerial photographs collected by the Canada
Department of Energy, Mines and Resources followed in August of 1959. These photographs show
prominent, exposed surface dirt layers and stratigraphic layering on the St. Patrick Bay ice caps, and the
Murray and Simmons ice caps are also bare of snow.  From digitizing the 1959 photographs and
mapping the ice cap outlines , the larger of the St. Patrick Bay ice caps then had an area of 7.48 $km^2$, and
the smaller one 2.94 $km^2$. The Murray and Simmons ice caps covered, respectively 4.37 and 7.45 $km^2$
(Serreze, 1985; Bradley and Serreze, 1987; Braun et al., 2004).  We estimate that these areas are
accurate to within 5%.
In July of 1972, Canadian scientists H. Serson and J. A. Morrison surveyed the larger of the two St.
Patrick Bay ice caps. They landed by helicopter in foul weather to find the ice cap totally covered with
snow. They installed eight accumulation stakes along a roughly two kilometer transect partway across
the ice cap.  The range in elevation along this transect was about 60 m, which compares to a range for
the entire ice cap of about 160 m.  Later that same summer, on August 20-21, the ice cap was visited by
G. Hattersley-Smith and A. Davidson, who noted a "partial cover of winter snow all around the ice
margins for at least a kilometer" (Hattersley-Smith and Serson, 1973), in striking contrast with
conditions depicted in the August 1947 and 1959 aerial photographs.  They concluded that while the ice
cap had been in decline (as suggested from the 1947 and 1959 photographs), by the early 1970s it had
returned to good health, "thickening slightly and extending its margins" (icy firn was observed atop the
dirty melt surface and a perennial snow cover extended beyond the ice cap margins).  This is consistent
with a known shift towards cooler summers and increased precipitation over the eastern Canadian
Arctic (Bradley and Miller, 1972; Bradley and England, 1978).  Hattersley Smith and Serson estimated a
mass balance for the 1971/1972 season of +0.14 m w.e.
In 1982 and 1983, the St. Patrick Bay ice caps were the focus of detailed energy and mass balance
investigations (Serreze, 1985; Bradley and Serreze, 1987; Serreze and Bradley, 1987).  The stake network
was expanded on the larger St. Patrick Bay ice cap and several stakes were installed on the smaller one.
At the end of the 1982 field season in early August, the entire ice cap was bare ice with a well-developed
cryoconite surface.  Assuming that the 1982 melt season had largely ended by early August (all visible
melt had stopped by the time that the field camp had been evacuated), the 1981/1982 mass balance for
the larger ice cap was estimated at -0.14 m w.e..  Given that more melt may have occurred, this is likely
a minimum estimate.  Based on the stake line installed in 1972, Bradley and Serreze (1987) estimated
that the overall mass balance for the period 1972-1982 was approximately -1.3 m w.e. (-0.14 a$^{-1}$).  This
result finds qualitative support in comparisons between the 1959 aerial photographs and subsequent
vertical aerial photographs taken on 1 August 1978 showing that the larger and smaller of the ice caps
had decreased  in area by 7% and 11% over that interval.  Like the 1959 photographs, the August 1978
photographs revealed a snow-free plateau and bare ice with a prominent ablation surface.  Aerial
photographs taken four years earlier, on 4 August 1974, showed broadly similar conditions.  As part of
the St. Patrick Bay Project, a network of stakes installed on the Simmons ice cap in 1976 (Bradley and
England, 1977) was re-surveyed on 11 July 1983.  Of the 18 original stakes, only 6 could be located; the
others were presumed to have melted out.  Based on these sparse data, Bradley and Serreze (1987)
estimated that over the period 1976-1983, the Simmons ice cap experienced a total mass loss of at least
-0.49 meters w.e. (-0.08 a$^{-1}$).  Collectively, these observations provided strong evidence that the period
of recovery inferred  by Hattersley-Smith and Serson (1973) was short-lived.
However, the summer of 1983 was cool, and the snow never completely melted off the surrounding
tundra.  The 1982/1983 annual mass balance for the larger St. Patrick Bay ice cap was estimated at
+0.14 m w.e. , and given their higher elevation, it is reasonable to assume that the 1982/1983 balance
year for the Simmons and Murray ice caps was also positive.

Table 1. Directly measured mass balances (meter water equivalent) of the Hazen Plateau ice caps. Where a value represents a multiyear record, the average annual value is shown in parentheses. Asterisks denote minimum estimates.

| Balance year or period | Large St. Patrick Bay | Small St.  Patrick Bay | Murray | Simmons |
|---|---|---|---|---|
| 1971/72 | +0.14[1] | ----- | ----- | ----- |
| 1971/72-1981/82 | -1.3[2] (-0.14)[2] | ----- | ----- | ----- |
| 1975/76-1982/83 | ----- | ----- | ----- | *-0.49 (-0.08)[2] |
| 1981/82 | *-0.14[2] | ----- | ----- | ----- |
| 1982/83 | +0.14[2] | ----- | ----- | ----- |
| 1983/84-1997/98 | ----- | ----- | ----- | *--0.49 (-0.03)[4] |
| 1983/84-1999/00 | *-1.01 (-0.06)[3] | *-1.26[3] (-0.07)[3] | ----- | ----- |
| 1998/99 | ----- | ----- | -0.49[3] | ----- |
| 1999/00 | ----- | ------ | -0.29[3] | -0.40[3] |
| 2000/01 | ----- | ----- | -0.47[3] | -0.52[3] |
| 2001/02 | ----- | ----- | -0.29[3] | ----- |
| Sources: [1]Hattersley-Smith and Serson (1973); [2]Bradley and Serreze (1987); [3]Braun et al. (2004) | | | | |


To our knowledge, there were no further visits to the Hazen Plateau ice caps until 1999, when C. Braun,
D. Hardy and R. Bradley of the University of Massachusetts Amherst established a network of 11
accumulation stakes on the Murray Ice Cap, which they further expanded in the year 2000.  A new
network of 15 stakes was established on the Simmons ice cap in 2000.  Winter snow accumulation was
measured on both ice caps in late May of 1999 through 2001, and summer ablation was measured in
late July and early August from 1999-2002.  For the four years analyzed, 1999-2002, annual balances of
both ice caps were negative in all years,  ranging from -0.29 m w.e. (Murray ice cap in 2000) to -0.52 m
w.e. (Simmons ice cap in 2001). In the summer of 2001, C. Braun and D. Hardy used portable GPS to
survey the perimeter of all four ice caps. The larger and smaller of the St. Patrick Bay ice caps had shrunk
to 62% and 59% of their 1959 areas, respectively. The Murray and Simmons ice caps had shrunk to 70%
and 53% of their 1959  areas.  Some of the accumulation stakes inserted into the larger St. Patrick Bay
ice cap in 1982 and 1983 were located but all had melted out.  Knowing how deep they had been
originally inserted enabled a minimum estimate (-1.01 m  w.e., -0.06 $a^{-1}$) of the mass loss between 1984
and 2000.  This is based on the mean remaining depth of stake insertion into the ice in 1983 and an
assumed ice density of 900 kg $m^3$ (Braun et al., 2004).  In the late summer of 2003, C. Braun mapped the
margins of the Murray and Simmons ice caps via portable GPS by holding the device out the window of a
low-flying helicopter.  The same approach was used to assess the ice cap margins in 2006, this time by
University of Massachusetts graduate student T. Cook.

### 3. Updated History, 1959 to 2016

3.1 Ice Cap Areas
The use of ASTER in conjunction with the air photographs and GPS surveys enables a fairly detailed
assessment of changes in ice cap areas from 1959 through the present.  Clear-sky late summer (July or
August) scenes of the St. Patrick Bay ice caps showing a strong brightness contrast between the ice and
the bare, dark plateau surface, enabled manual mapping of the ice cap perimeters from ASTER for the
years 2005, 2009, 2014, 2015 and 2016.  For the Murray and Simmons ice caps, ASTER estimates were
obtained for 2001, 2007 and 2016. For 2001, areas of the Murray and Simmons were available from
both ASTER and the surface-based GPS surveys.  Considering the GPS surveys for this year as ground
truth, the ASTER areas for this year are accurate to within 1% for the Murray ice cap and 3% for the
Simmons ice cap.  It is assumed that this is representative of the accuracy of area mapping from ASTER
for the other years.
As of July 2016, and Murray and Simmons ice caps cover 39% and 25% of the areas in 1959 based on the
aerial photographs.  By sharp contrast, both of the St. Patrick Bay ice caps in 2016 cover only 5% of their
former areas, and have been reduced to ice patches, with the smaller ice body now covering only  0.15
$km^2$.

| Table 2. Surface areas ($km^2$) and % areas compared to 1959 aerial photographs | | | | | | | |
|---|---|---|---|---|---|---|---|
| | Larger St. Patrick Bay Ice Cap Area and % of 1959 | | Smaller St. Patrick Bay Ice Cap Area and % of 1959 | | Murray Ice Cap Area and % of 1959 | | Simmons Ice Cap Area and % of 1959 |
| 1959 | 7.48[1] | 100% | 2.94[1] | 100% | 4.37[1] | 100% | 7.45[1] | 100% |
| 1978 | 6.69[1] | 89% | 2.74[1] | 93% | ------ | ------ | ------ | ------ |
| 1999 | ------ | ------ | ------ | ------ | 3.28[2] | 75% | ------ | ------ |
| 2000 | ------ | ------ | ------ | ------ | 3.15[2] | 72% | ------ | ------ |

| 2001 | 4.61[2] | 62% | 1.72[2] | 58% | 3.05[2](3.08[4]) | 70% | 3.94[2](3.83[4]) | 53% |
|---|---|---|---|---|---|---|---|---|
| 2003 | ------ | ------ | ------ | ------ | 2.91[3] | 66% | 3.31[3] | 44% |
| 2005 | 3.68[4] | 49% | 1.03[4] | 35% | ------ | ------ | ------ | ------ |
| 2006 | ------ | ------ | ------ | ------ | 2.86[3] | 65% | 3.19[3] | 43% |
| 2007 | ------ | ------ | ------ | ------ | 2.76[4] | 63% | 2.92[4] | 39% |
| 2009 | 2.54[4] | 34% | 0.63[4] | 21% | ------ | ------ | ------ | ------ |
| 2014 | 1.03[4] | 14% | 0.29[4] | 10% | ------ | ------ | ------ | ------ |
| 2015 | 0.52[4] | 7% | 0.18[4] | 6% | ------ | ------ | ------ | ------ |
| 2016 | 0.35[4] | 5% | 0.15[4] | 5% | 1.72[4] | 39% | 1.84[4] | 25% |
| [1]Aerial photographs, [2]Surface GPS surveys, *Braun et al.* (2004), [3]GPS helicopter surveys, [4]ASTER | | | | | | | | |


Outlines of the St. Patrick Bay ice caps for 1959 from aerial photography, for 2001 from GPS surveys, and
for 2014 and 2015 from ASTER are shown in **Figure 2**.  The reductions in ice cap area are striking.  Note
the obvious shrinkage even between the years 2014 and 2015.  Shrinkage of the Murray and Simmons
ice caps is shown in **Figure 3**, based on outlines from 1959, 2001 and 2016.  The shrinkage of these two
ice caps is clearly evident, albeit less pronounced.
Using the area estimates through 2002 and extrapolating forward, Braun et al. (2004) suggested that the
Hazen Plateau ice caps would disappear by the middle of the 21$^{st}$ century or soon thereafter, and that,
given their larger size, the Simmons ice cap and the larger of the two St. Patrick Bay ice caps would be
the last to go.  However, based on data through 2016 and extrapolating forward (**Figure 4**), it now
appears that both of the St. Patrick Bay ice caps will disappear around the year 2020.
From the analyses described above, and results from other glaciological investigations for the Canadian
Arctic and the Arctic as a whole, the following conclusions are drawn:
• The Hazen Plateau ice caps are unlikely to be relics of the last glacial maximum, but rather likely
formed during the Little Ice Age (LIA, c. 1600-1850) (Koerner, 1989).  They may have retained
their LIA extents through the first couple decades of the 20$^{th}$ century (Hattersley-Smith, 1969;
Sharp et al., 2014), but have been in overall decline ever since.  Braun et al. (2004) speculate on
the basis of a mapped lichen trim line that the Murray ice cap may have attained a maximum LIA
extent of about 9.6 km$^2$, over twice the mapped 1959 area of 4.35 km$^2$.  Similar trim lines were
observed around the other three ice caps and although not mapped in detail, strongly point to
much more extensive ice cover during the LIA. To place these findings in a broader context, for
the Queen Elizabeth Islands as a whole, trim lines based on high-resolution satellite imagery
point to a 37% reduction in perennial snow and ice extent between the LIA maximum extent and
the year 1960. Over the lower-lying central and western islands, a complete removal of
perennial snow and ice occurred by 1960 (Wolken et al., 2008).
• From the 1960s through part of the 1970s, the ice caps may have experienced a period of
reduced loss or occasional growth (1971/1972, 1982/1983) in response to cooling.  This basic
pattern likely holds for monitored Canadian Arctic glaciers and ice caps as a whole (Bradley and

Miller, 1972; Hattersley-Smith and Serson, 1973; Ommanney, 1977; Bradley and England, 1978;

Braun et al., 2004; Sharp et al., 2014).

•   Since then, apart from occasional years such as 1982/1983, annual mass balances of the four ice

caps have been persistently negative (Braun et al., 2004).  This is in turn consistent with the

broader pattern of reductions in mass and area of Arctic glaciers and ice caps (Dowdeswell  et

al., 1997; Dyurgerov and Meier, 1997; Arendt et al., 2002; Koerner, 2005; Sharp et al., 2011,

2014; Fisher et al., 2012; Sharp et al., 2014; Mortimer et al., 2016).  It is also consistent with a

negative mass balance of the Greenland Ice Sheet since at least the 1990s (Shepard et al., 2012).

Mass balance summaries for four monitored glaciers and ice caps in the Canadian Arctic (Devon Ice
Cap, Meighan Ice Cap, Melville South Ice Cap and the White Glacier) are provided as part of the
American Meteorological Society (AMS) State of the Climate reports.  As assessed over the period
1980 through 2010, all four have had negative average annual mass balances, ranging from -0.15 m
w.e. for the Devon Ice Cap to -0.29 w.e. for the Melville South Ice Cap (AMS, 2016).  Cumulative
changes in regional total stored water for the period 2003 through 2015 based on gravimetric data
from the GRACE mission (Gravity Recovery and Climate Experiment) are qualitatively consistent with
these mass balance measurements (AMS, 2016). Based on ice core data, Fisher et al. (2012)
document rapid acceleration of ice cap melt rates of over the last few decades across the entire
Canadian Arctic; the large reductions in area of the Hazen Plateau ice caps, in particular the lower-
elevation St. Patrick Bay ice caps, is consistent with this finding.  However, reflecting variable climate
conditions, annual balances are also quite variable.  For example, for the 2013/2014 balance year
(the most recent data available), the White Glacier had a strongly negative balance (-0.42 m w.e.)
while the small Meighan Ice Cap actually gained mass (+0.06) (AMS, 2016).  Sharp et al. (2014) show
that while the larger ice bodies in the Canadian Arctic have seen the largest losses in mass, the
smaller masses have lost a larger proportion on their areas.  This is also consistent with the behavior
of the Hazen Plateau ice caps.  Below we examine variability in climate conditions over the Hazen
Plateau, and links to mass balance and area changes.
3.2 Associated Climate Conditions
The annual mass balance of low-accumulation ice caps and glaciers in the Canadian High Arctic is known
to be primarily governed by summer warmth rather than winter accumulation (e.g., Bradley and
England, 1978; Koerner, 2005).  To place the behavior of the Hazen Plateau ice caps in a climate context,
use is made of summer-averaged (June through August) 850 hectopascal (hPa) temperature anomalies
from the radiosonde record at Alert, located on the northeast coast of Ellesmere Island  (Figure 1) along
with estimated summer temperature anomalies for the LIA. The Alert radiosonde record extends back
to 1957.  We use monthly mean records contained in the Integrated Global Radiosonde Archive (IGRA,
Durre et al., 2006), based on daily 00 and 12 UTC soundings.  Summer averages (J,J,A) were eliminated if
based on fewer than 70 values.  The 850 hPa level is about 1400 m above sea level for a standard
atmosphere, hence roughly 600-700 m above the surface in the vicinity of the St. Patrick Bay ice caps
and 300-400 m above the surface in the vicinity of the Murray and Simmons ice caps.  While arguably it
might be better to look at the 925 hPa level as it is closer to the plateau surface, this level has many
missing values in the IGRA record.  The time series of anomalies, computed with respect to the standard
averaging period 1981-2010, follows in **Figure 5**.
Kaufman et al. (2010) took advantage of a variety of proxy sources (e.g., tree rings, ice cores, lake cores)
to assemble a record of Arctic summer surface temperature anomalies that extend back 2000 years.
From their analysis, LIA summer Arctic temperatures anomalies averaged around -0.6$^o$C with respect to
a 1961-1990 reference period.  The 1961-1990 summer mean of -3.2$^o$C from the radiosonde data
compares to a mean of -2.6$^o$C for 1981-2000.  The latter period is hence about 0.6$^o$C warmer.  With the
assumption that (a) temperature anomalies at 850 hPa  have been similar to those at the surface
(supported by Sharp et al. (2011) in their analysis of Canadian Arctic ice caps but possibly complicated by
the temperature inversion structure), and (b) LIA conditions over the Hazen Plateau were at least
broadly similar to those for the Arctic as a whole, these results imply that Arctic LIA temperature
anomalies were about -1.2$^o$C relative to a 1981-2000 baseline.  This estimated LIA temperature anomaly
is shown in Figure 5 as a dashed line.
If it is also accepted that the ice caps were broadly in equilibrium with average LIA summer
temperatures, Figure 5 suggests generally strong negative annual balances from the beginning of the
record through the early 1960s.  This was followed by smaller negative and occasionally positive annual
balances from the middle of the 1960s through about 2000, and a preponderance of strong negative
balances from the beginning of the century through the present.  For comparison with the radiosonde
record, we also examined 850 hPa summer temperatures over the Hazen Plateau from the National
Centers for Environmental Prediction/National Center for Atmospheric Research (NCEP/NCAR)
reanalysis (Kalnay et al., 1996) which extend back to 1948.  Given that the Alert radiosonde data are
assimilated into the reanalysis, it follows that the radiosonde and NCEP/NCAR time series look similar
for the period of overlap.  The NCEP/NCAR records suggest that the period 1948-1956 not covered by
the IGRA record was warm overall  with mostly positive anomalies relative to the 1981-2000 baseline.
The cooling between the late 1940s through the middle 1960s broadly corresponds to the cooling over
the eastern Canadian Arctic such as discussed by Bradley and Miller (1972), Bradley and England (1978)
and other studies. The time series of decadal mean summer temperatures  at the 700 hPa level for the
major glaciated regions of the Canadian Arctic presented by Sharp et al. (2011) based on the
NCEP/NCAR reanalysis (their Figure 9.3) is also consistent with the pattern shown in Figure 5.
An examination of selected individual years is instructive. Based on summer 1957 temperatures, the
1956/1957 annual balance must have been strongly negative.  The same can be said for 1959/1960 and
1962/1963. By sharp contrast, the summer of 1972, when Hattersley-Smith and Serson (1973) visited
the ice caps and remarked upon the extensive August snow cover over the plateau and estimated a
positive balance of +0.14 m w.e. (for the 1971/1972 season), was the coldest in the radiosonde record,
and about 2$^o$C below the estimated LIA average.  There is also a clear contrast between 1982 (a known
negative annual balance year for the St. Patrick Bay ice caps) and 1983 (a known positive balance year,
with summer 850 hPa temperatures slightly below the LIA average).  Given the low temperatures for the
summer of 1992, which followed the 1991 eruption of Mt. Pinatubo, the balance for 1991/1992 was
likely positive.  In the sense that summer temperatures were above the estimate LIA average, negative
balances for the Murray ice caps for 1998/1999 through 2001/2002 (Braun et al., 2004) are all
consistent with Figure 5.  Note however that the largest negative balance of -0.49 m w.e. for 1999/2000
corresponds to the coldest of the four summers, arguing for influences of local effects on summer
temperature or perhaps a low winter accumulation.
Regarding the summer of 2013, the obvious exception to the pattern of recent warm years, the ASTER
data and daily images from the Moderate Resolution Imaging Spectroradiometer (MODIS) show
extensive cloud cover through the summer, making it difficult to determine whether the snow cover
ever entirely cleared off the plateau.  It is likely, however, that the 2012/2013 balance year was positive
for the Hazen Plateau ice caps - the Devon Ice Cap, Meighan Ice Cap and the White Glacier all gained
mass. Only the Melville South Ice Cap, lying well to the west, had a negative balance (AMS, 2014).
Consistent with this view, **Figure 6** shows that summer (J,J,A) averaged 850 hPa temperature anomalies
over the Queen Elizabeth Islands from the NCEP/NCAR reanalysis were about $2^o$C below the 1981-2010
baseline in the area centered over Axel Heiberg and Ellesmere islands.  This reflects the influence of an
unusually-deep circumpolar vortex at the 500 hPa level, centered just south of the Pole along about
$90^o$W longitude.  By sharp contrast, the notable area reduction of the St. Patrick Bay ice caps between
August 2014 and 2015 aligns with the very warm summer of 2015, essentially tied with 1957 as the
highest in the record.  From **Figure 7**, July 2015 temperatures at the 850 hPa level from the NCEP/NCAR
reanalysis were 3-4$^o$C above the  1981-2010 baseline over most of northeastern Ellesmere Island.  Mass
balance estimates for monitored glaciers in the Queen Elizabeth Islands for the 2014/2015 season that
would provide context were not available us at the time that this paper came to press.

**4. Conclusions**

Regarding accelerating wastage of the St. Patrick Bay ice caps since the dawn of the 21$^{st}$ century, the
outsized warming of the Arctic in recent decades compared to the rest of the Northern Hemisphere
(termed Arctic Amplification), is overall most strongly expressed during the cold season, and is not
nearly as prominent in summer (Serreze and Barry, 2011). Nevertheless, from the NASA Goddard
Institute for Space Sciences (GISS) analysis (http://data.giss.nasa.gov/gistemp/), the trend in July surface
air temperatures over Northeastern Ellesmere Island over the period 1960-2015 is about 2$^o$C (expressed
as a total change) which stands out compared to the rest of the Arctic.  On the basis of a satellite-
derived record (from MODIS) of summer land surface temperatures, the more recent period of 2000 to
2015 has seen an average warming rate over the Queen Elizabeth Islands of 0.06$^o$C per year, or a total of
nearly 1.0$^o$C, most of this occurring between 2005 and 2012 (Mortimer et al., 2016).  They associate this
warming with increasingly negative mass balances for glaciers and ice caps in the region.  However,
conditions over the Hazen Plateau are highly variable, and the summers of 1957, 1960 and 1963 were
almost as warm as those seen in 2015, and the summer of 2013 was quite cool, very likely resulting in a
positive balance for 2012/2013.
Rapid wastage of the St. Patrick Bay ice caps over the past 15 years likely also reflects a reduction in
summer albedo, as dirt layers become progressively exposed and accumulate at the surface.  During the
1982 and 1983 field campaigns, it was observed that summer precipitation over the ice caps was
typically in the form of snow, temporarily increasing the surface albedo and adding some mass.  The
frequency of summer snowfall has likely declined in the (generally) sharply warming climate over the
past 15 years.  Also, as suggested from the prominent decline in the area of the larger St. Patrick Bay ice
cap between 2014 and 2015, when there is an especially warm summer, the thin collar of ice at the ice
cap margins (a feature evident in field observations) will be prone to completely melting.  The less
pronounced area reduction of the Murray and Simmons ice caps must partly be due to their higher
elevation and relatively cooler summer conditions.  However, the elevation difference is only about 200-
300 m, which argues that the stronger response of the St. Patrick Bay ice caps to warming may also be
related to ice thickness.   Regional differences in the temperature lapse rate (notably the temperature
inversion structure) could also be involved.
It is possible that the Hazen Plateau caps could see some temporary recovery given the large natural
variability in the Arctic.  However, as noted by Alt (1978) and Bradley and England (1978), for stagnant
ice caps such as these, all it takes is one warm summer to erase any accumulated mass gains of a
previous decade.  Assessing variability and trends in Arctic precipitation is notoriously difficult, but as
evaluated over the period 1950-2007, annual precipitation has generally increased across Canada, and
especially across Northern Canada.  For example, at station Eureka in central Ellesmere Island (see
Figure 1), annual precipitation appears to have increased by at least 40% (Zhang et al., 2008).  Trends
over the plateau are not known, but this suggests that, if anything, precipitation changes are helping to
buffer the ice caps from summer mass loss.
Paradoxically, perhaps, loss of the Hazen Plateau ice caps may open new research opportunities. As they
recede, plant remains are exposed that can be dated and used to better understand the past climate
history of the region. From radiocarbon dates on rooted tundra plants exposed by receding cold-based
ice caps on Baffin Island – given that the plants are killed when the snowline drops below the collection
sites – Miller et al. (2013) were able to construct a record of summer temperatures over Arctic Canada
for the past 5000 years.  La Farge et al. (2013) discovered that ice loss in Sverdrup Pass, Ellesmere Island,
has exposed nearly intact plant communities for which radiocarbon dates point to entombment during
the LIA.  They also found that these recently-exposed, subglacial bryophytes can regenerate, which may
have important implications for recolonization of polar landscapes.  The area surrounding the receding
Hazen Plateau ice caps provides a unique opportunity to examine this process of recolonization in the
High Arctic, as the rates of ice recession are now well-documented for the last 55+ years (Table 1).
**Author Contribution:** M. Serreze led the overall effort.  C. Braun, D. Hardy, and R.S. Bradley provided
GPS data and historical documents.  B. Raup analyzed the ASTER data.  All authors contributed to the
writing.
**Data Availability**:  Radiosonde data for station Alert are available at from the Integrated Global
Radiosonde Archive (ftp://ftp.ncdc.noaa.gov/pub/data/igra/).  ASTER data can be obtained through the
NASA Land Processes DAAC (https://lpdaac.usgs.gov/).
**Acknowledgements:**  This study was supported by the University of Colorado Boulder, the NASA Snow
and Ice DAAC award NNG13HQ033 to the University of Colorado, and NSF Award 9819362 to the
University of Massachusetts.
**Competing Interests:** None

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

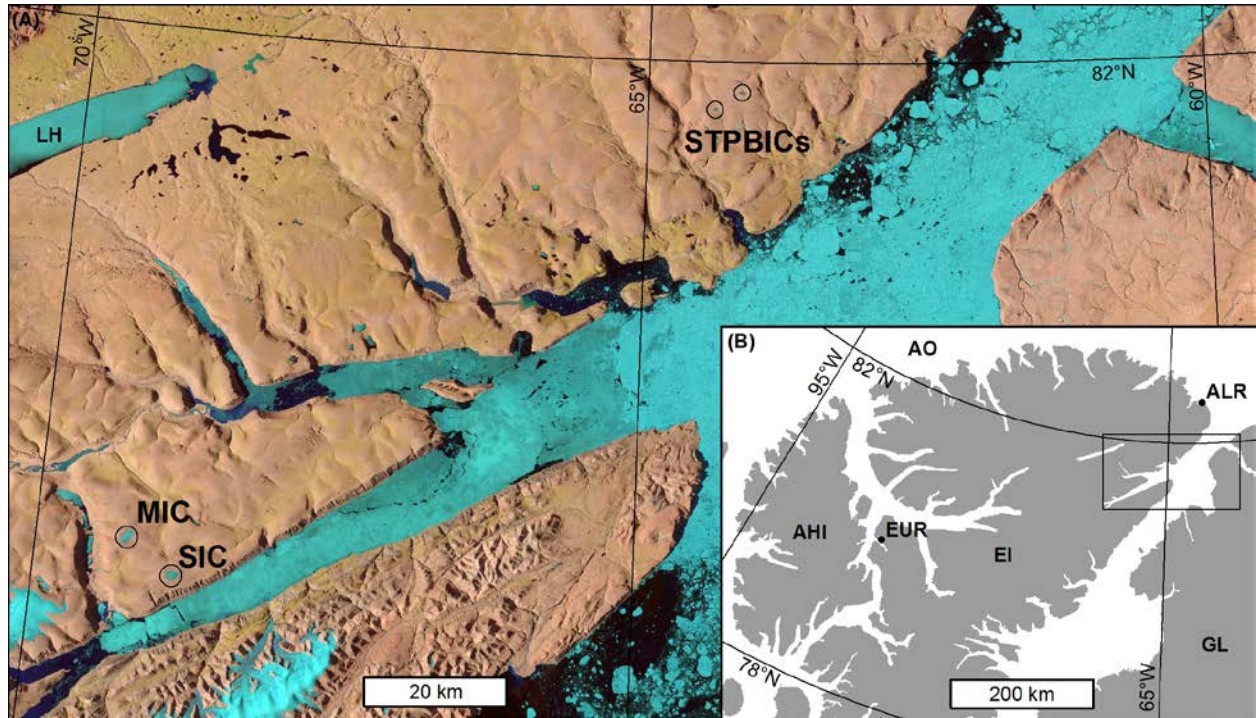


Figure 1. The location of the St. Patrick Bay (STPBIC), Murray (MIC) and Simmons (SIC) ice caps.  The
inset map shows Ellesmere Island (EL), Axel Heiberg Island (AHI), Greenland (GL), the Arctic Ocean (AO)
and station Alerts (ALR) and Eureka (EUR).  Use is made of 850 hPa temperature data from the Alert
radiosonde record and precipitation records from Eureka.

434

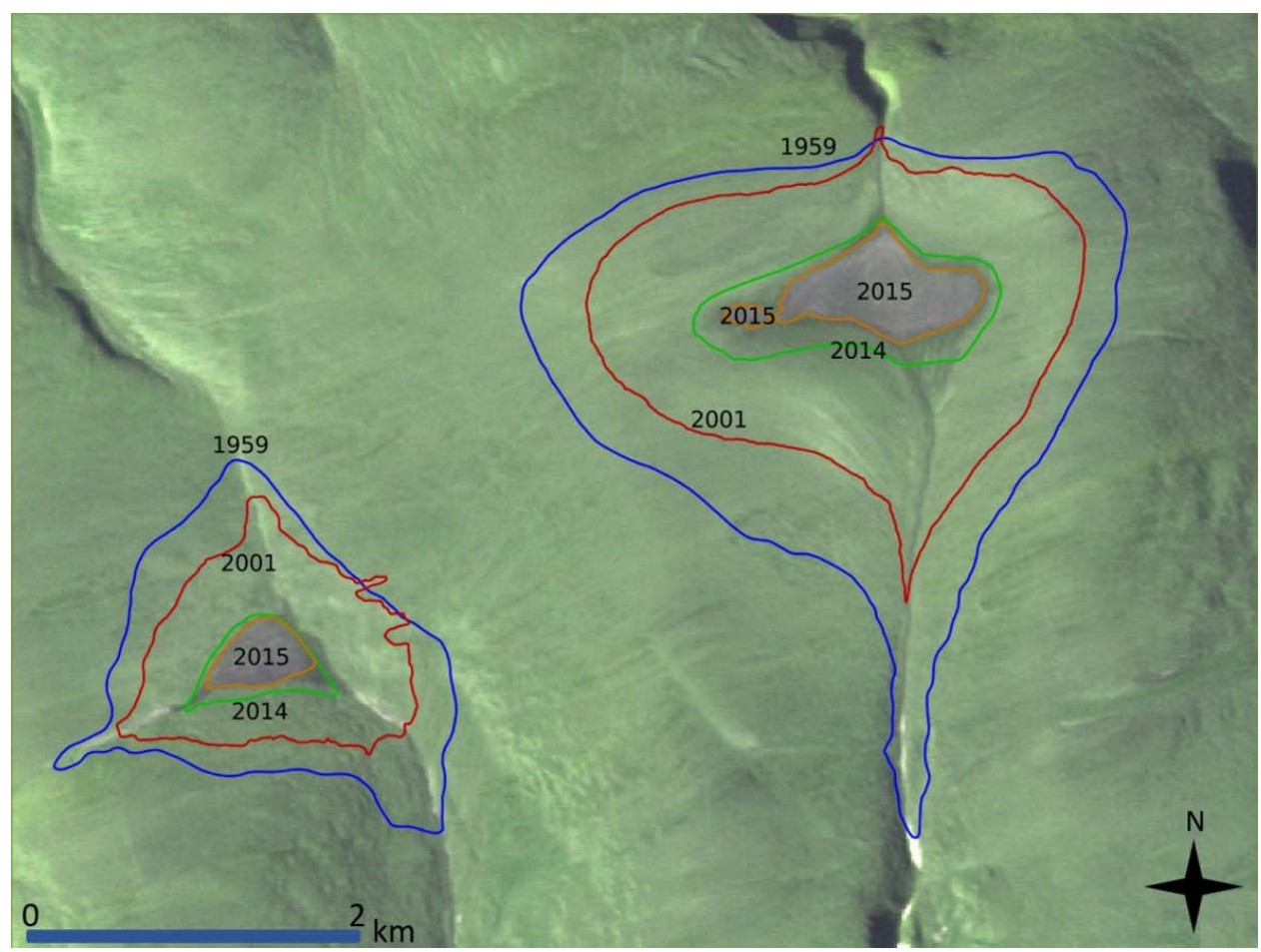

Figure 2:  Outlines of the St. Patrick Bay ice caps based on aerial photography from August 1959, GPS surveys conducted during August 2001, and for August of 2014 and 2015 from ASTER.  The base image is from August 2015.

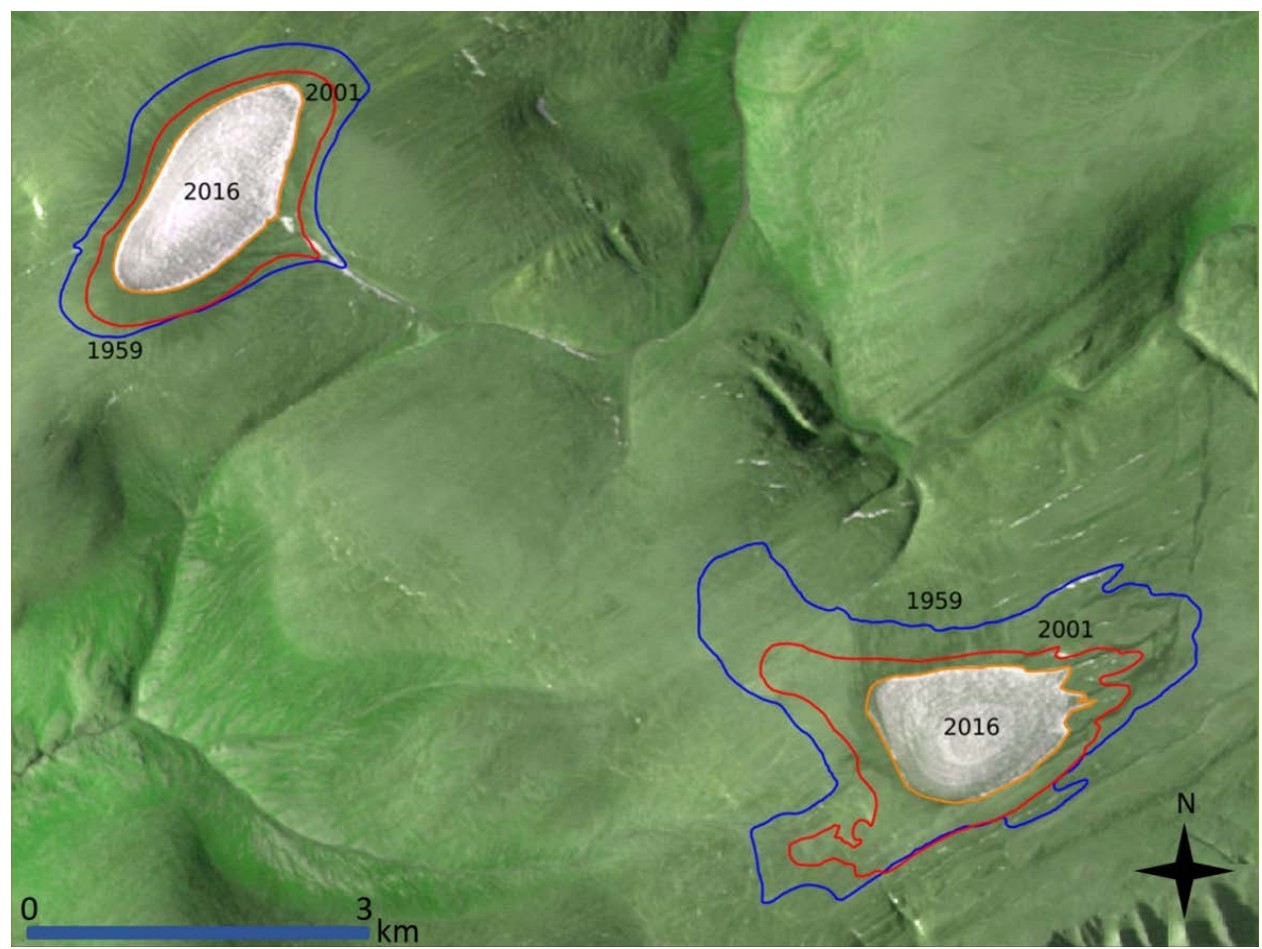

Figure 3: Outlines of the Murray and Simmons ice caps based on aerial photography from August 1959, GPS surveys conducted during August 2001, and for July 2016 from ASTER. The base image is from August 2016.


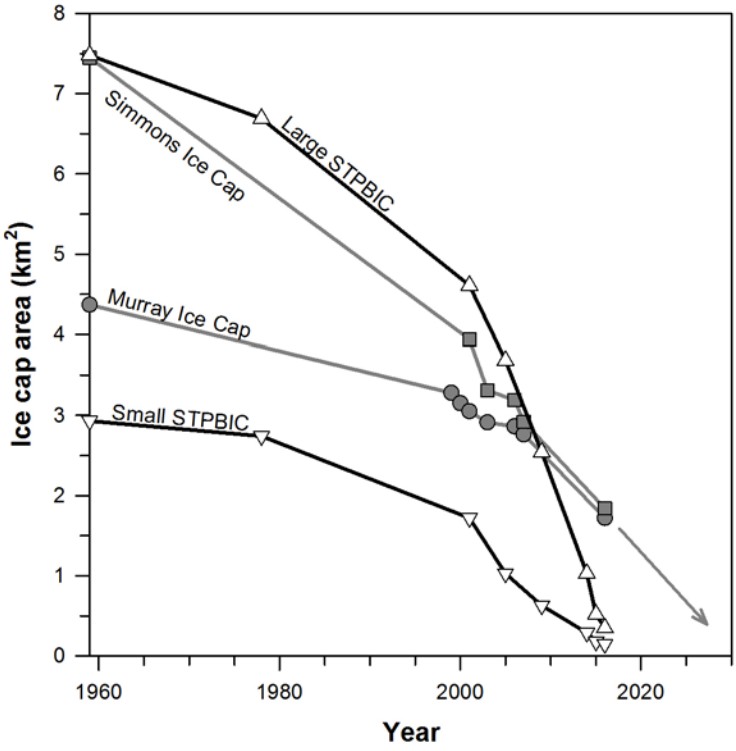


Figure 4.  Time history of ice cap areas and projected times of disappearance.

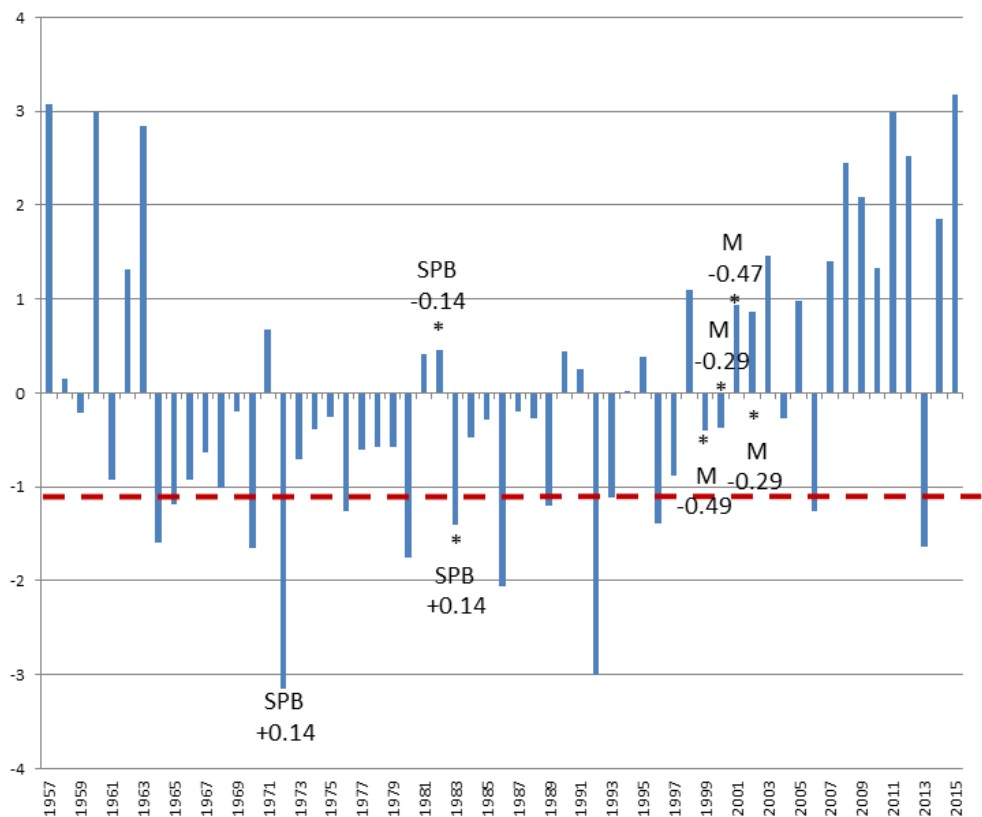


Figure 5. Temperature anomalies at the 850 hPa level ($^o$C) over the period 1957-2016 (referenced to the
period 1981-2010) from the Alert radiosonde record.  The dashed red line shows the estimated summer
average Arctic temperature anomaly for the LIA relative to 1981-2010.  Also shown are the annual mass
balance estimates for the larger St. Patrick Bay (SBP) ice cap for the 1981/1982 and 1982/1983 balance
years, and for the Murray Ice Cap (M) for the 1999/2000, 2000/2001, 2001/2002 and 2002/2003
balance years (in m w.e.).

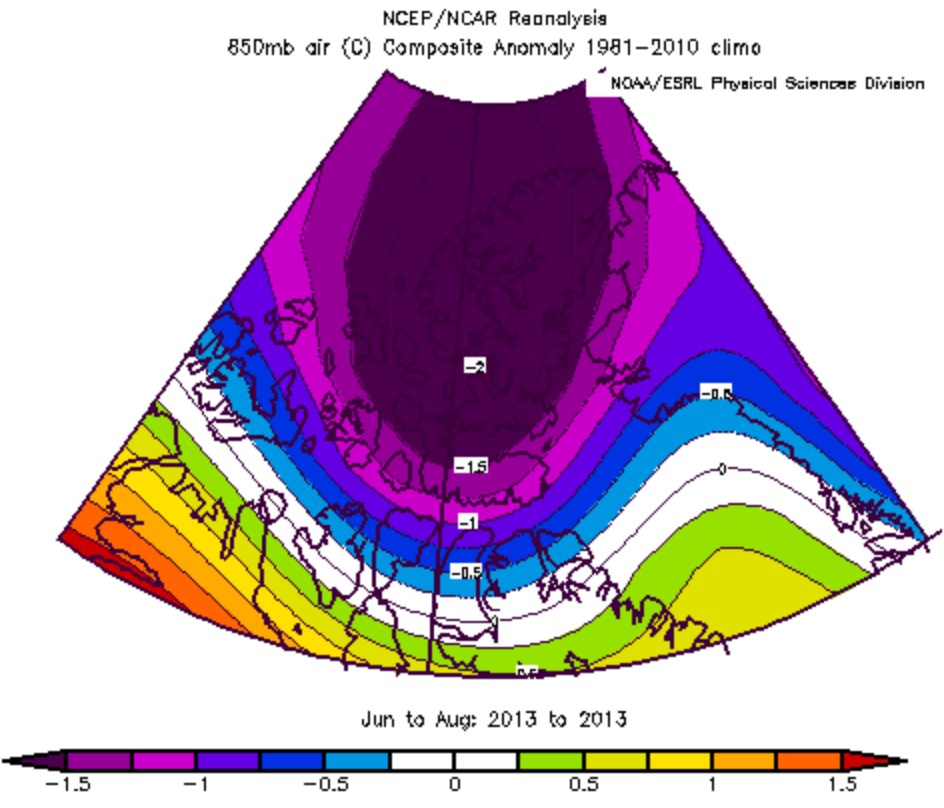



Figure 6. Summer  (J,J,A)  2013 air temperature anomalies at the 850 hPa level from the NCEP/NCAR
reanalysis relative to a 1981-2010 baseline.

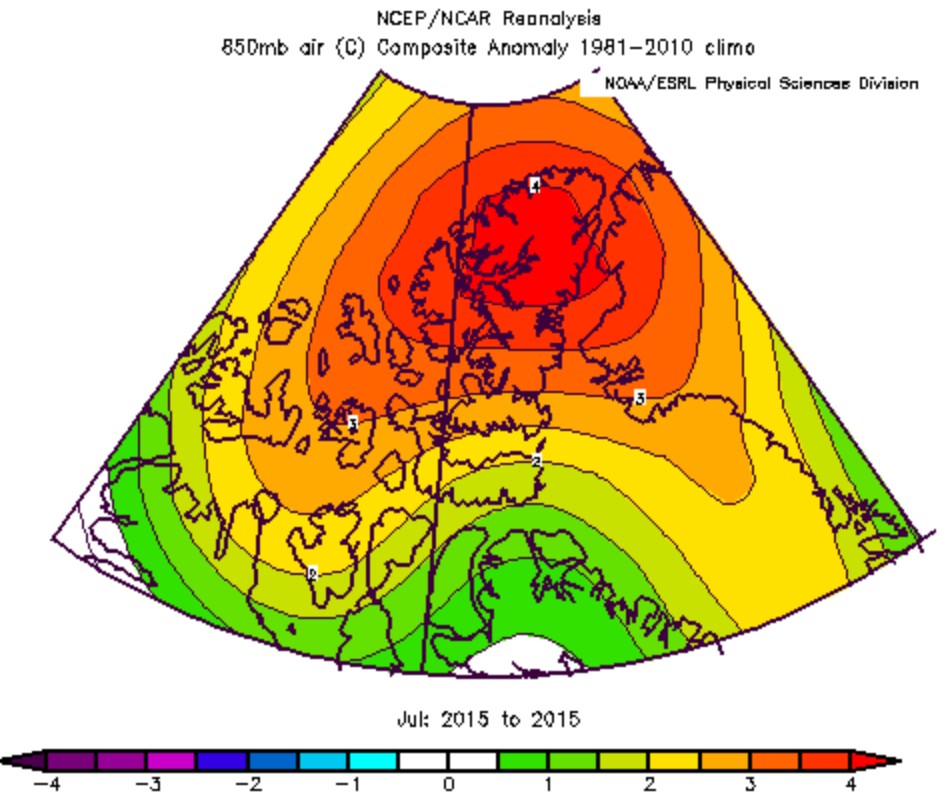



Figure 7. July 2015 air temperature anomalies at the 850 hPa level from the NCEP/NCAR reanalysis
relative to a 1981-2010 baseline.



