# Peer review of "Rapid Wastage of the Hazen Plateau Ice Caps, Northeastern Ellesmere Island, Nunavut, Canada"

_The Cryosphere, 2016_

## Referee Comment (RC1) · Anonymous Referee #1 · 5 Oct 2016

This paper documents the near demise of 4 small ice caps in NE Ellesmere Island - ice caps that have a >50 year history of study. The paper is of some interest because of this long history, but it could be made more interesting if more effort were made to place the results in a broader regional context.

Specifically, there is no mention at all of the work of Gabriel Wolken, who used trimline mapping to document the pattern of post Little Ice Age glacier retreat across the Queen Elizabeth Islands and interpret observed patterns in terms of past climate dynamics (The Holocene 18 (4), 615-628 and 629-641, 2008). Nor is there mention of the chapter on this region in the GLIMS book (edited by Jeff Kargel and others; Global Land Ice Measurements from Space, pp 205-228 (Springer, 2014)), which provides a sub-regional breakdown of post 1950's ice retreat across the area as a function of initial ice cap/glacier size. Many small ice caps have disappeared from this region in the

past 60 years, but this would not be obvious from reading the submitted manuscript).

Equally, there is no reference to the 4 in situ mass balance time series from the area which provide a nice picture of the short term variability in climate-mass balance linkages that would help with the interpretation of the results presented here - or of the regional ice mass change time series from GRACE, which would do the same thing. The relevant data are published annually in the Arctic section of the BAMS "State of the Climate" report and are readily available (as are the mass balance time series from WGMS). Nor is any comparison with ice core records from the region, which would also help to provide longer term context for the work (e.g. Fisher et al., 2012, Global and Planetary Change 84, 3-7).

As a result, the paper seems somewhat disconnected from what is already known about post LIA and recent glacier change in the region and its drivers. I think this issue has to be addressed if the paper is to pass the "significance" test for publication in The Cryosphere. The paper does nicely document the history of these specific ice caps in more detail than would be possible for most others in the same region (hence good for originality) , and the detail is sufficient to allow reasonably sophisticated comparison of ice cap retreat rates and patterns with other climate and mass balance indicators for the region - but this is not really attempted (hence fair for scientific quality and significance). This leaves the paper with a rather anecdotal feel. I think this needs to be changed before I could recommend publication in this journal.

The paper is however quite readable (good for presentation quality) although I do suggest below quite a number of detailed edits that would make it more readable.

Specific Points (keyed by line number):

71-74: Specify uncertainties associated with the ice cap area estimates - important to know how large these are relative to the observed changes

77: what is the range of surface elevations covered by this transect and how does it

compare with the total elevation range of the ice cap?

85-86: this statement seems like unnecessary speculation, given that the comparison is with the behaviour of a single studied ice cap.

91: Why assume the 1982 melt season ended by the end of August ? Evidence for this claim? Climate data?

92-94: an annual mean MB value for a given time interval might be more useful than a period mean and, if there is a stake line, it would be useful to say something about how the annual balance varies with elevation.

96: decreased in area (rather than shrunk in area)

103: Is -0.49 the annual value or a period mean? Not clear

104: inferred by Hattersley-Smith. . .

106-107: Treats temperature and mass balance as interchangeable terms - not justified, so talk about temperature as that is what the data relate to (or present a quantitative relationship that justifies inferring MB from T

109: 1982/83 balance of the Simmons.. . .

116-117: annual balances of both ice caps were negative in all years, ranging from. . .

120: The larger and smaller. . . ..of their 1959 areas respectively, while the Murray. . ...had shrunk to 70%. . . . . .their 1957 areas

123: inserted allowed us to make a minimum estimate (-1.10 m w.e.) of the mass loss. . .

125-128: given the measurement approach, I think some assessment of the associated errors is warranted.

129: . . ..studies, and the results from. . ..

131: maximum, and likely to have formed

133: in the first couple. . ..

134-135: On the basis of a mapped lichen tramline, Braun et al. (2004) speculate that

135-138: Here the authors should make reference to Wolken's study (The Holocene) of lichen trimlines in the QEI; I'd also like to see a tabulation of all the available surface mass balance measurements and the time periods they represent.

143: List all the known positive balance years so we can see how many there have been

147: Note there are GRACE mass balance time series for the QEI (see the Arctic section of successive annual BAMS "State of the Climate" reports) and for the Russian Arctic, Svalbard, and Alaska, so you could make broader scale Arctic comparisons with your data time series.

152: areas (in km2) from all..

157: of the Murray

162: 2016, the Murray. . .. . ...in 1959. By sharp contrast, . . .. . .. . .. . ...only 0.15 km2

168: reductions in. . .. . ...area are striking

170: is shown in Figure 3, based on outlines from 1959, 2001, and 2016.

177: Note that none of these studies discuss glacier area changes, and you don't reference the one that does (paper in the GLIMS book)

197: is it meaningful to make comparisons with pan-Arctic means given the scale of this study?

200-203: Sharp et al (2011) provide evidence that would support this assumption (i.e. that 850 hPa and surface air temperatures show similar patterns)

216: Arctic discussed by. . .

231: as the warmest in the record.

236: anomalous warming 237: is most strongly. . . 241: stands out how? 242: make it clear that these LST results relate to ice covered surfaces only

245-257: Do these specific years stand out as anomalous in the long term surface mass balance records from elsewhere in the QEI (White Glacier, Meighen Ice Cap, Melville South Ice Cap, Devon ice Cap) ? Why not discuss these data?

248: likely reflects. . . 251-252: Be careful here, MODIS data suggest that this assertion would not hold everywhere in the QEI 254: when there was an especially. . . . . .. . ..in field observations) 258-59: related to their limited thickness 266-268: note that MODIS albedo data do provide support for this hypothesis in restricted areas of the QEI 280: Do you have any specific information on what plant taxa have been exposed by the retreat of these ice caps?

---

## Referee Comment (RC2) · Anonymous Referee #2 · 12 Oct 2016

Overview: This paper documents the important and timely phenomenon of disappearing ice caps in the Canadian high arctic. A time series of area measurements is compiled primarily from previously published observations, with a small contribution of new measurements to document shrinkage and predict the timing of the demise of the plateau ice caps. Some previously published surface mass balance measurements are reported but not discussed. The area changes documented in this study are linked through qualitative comparison to annual temperature 850hPa radiosonde temperatures from Alert. A major shortcoming of this paper is the under-utilization of available long term records from other arctic glaciers to determine if the rate of glacier change over the hazen plateau is representative or anomalous of the broader scale glacier changes occurring in the Canadian high Arctic. A more rigorous quantitative analysis of the complete time series of changes to the Hazen plateau ice caps should be made

in order to contribute to the broader understanding of the rates of climate change in the Canadian high arctic and the current and future rate of contribution of ice caps and glaciers in this region to global sea-level rise.

Comments: L58-60: A major shortcoming of this paper is lack of new geophysical data generated for this study, as such, there is no needs for a methods section. The distinction between which data were produced by the authors for this paper, and data/information from previously published material needs to be more clear.

- While the authors illustrate the rate of shrinkage of the hazen plateau ice caps though plots of the time series, there is no attempt made to determine if the rapid changes as determined from this study are occurring at anomalous relative to those documented for other high arctic glaciers.

L49: It would be more informative if the actual elevations of the ice cap are rather than just the maximum surrounding area (ie. "...ice caps are in an area with maximum elevations s between 750-900 m;...") as stated.

Extent of LIA glacier cover (Wolken et al.) should be included in the analysis to provide a longer term perspective to the changes discussed in this study.

L53-54: the authors should clarify what form of precipitation this statement ("..., with a late summer and early autumn maximum..") refers to ie. Rain or snow, or a combination of both, as they can have opposite impacts on the mass balance of small ice caps with no firn to absorb liquid precip. L83: Presumably these ice caps are stagnant. However, the authors refer to the ice caps "...extending its margins,.." which may be misinterpreted as advancing via flow, which is almost certainly not the case. It is most likely that the" extended" margins are actually perennial snow packs which would be of lower density material than the original ice cap. This should be clarified L83: "...thickening slightly ..." how was 'thickening' determined? L91: "Assuming that the 1982 melt season had largely ended by early August..." unless there is temp data to support this claim, there is no reason to assume that the melt season ended in early

august. High arctic glaciers at these elevations commonly experience melting into late august.

L94 and 103: it is more informative (and more common in glaciology) to report mass balance as an annual (ie a-1) value even when measurements span multiple years. 192 " While arguably it might be better to look at the 925 hPa level," the authors need to explain why this is the case. L178/179: the studies referenced refer to loss of ice mass or surface mass balanace, not specifically area change. This is an important distinction (and should be discussed) because area reductions of the larger dynamic ice caps are also a function of dynamic response time whereas the margins of small plateau ice caps respond immediately to surface ablation and would shrink at faster rates relative to the dynamic ice masses. Figure 5: it would be helpful to integrate the annual and multi year average surface mass balance measurments and/or area change values from all studies (this one and all referenced herein) into fig 5 in order to show the relationship between temp change and ice cap response. L54: the serreze and barry 2015 is listed as 2014 in the refs. Figure 2. scale and north arrow unreadable – too small. Figure 1. need to indicate location of Environment Canada weather stations from which data is used. Alert is identified, but should be stated in the caption that it is one location of the long term temp data. Eureka (from which precip data is obtained) is not on the map at all.

---

## Author Comment (AC2) · 7 Nov 2016

Editor, The Cryosphere:

Responses to the comments and suggestions to the second anonymous reviewer of manuscript tc-2016-201 Rapid Wastage of the Hazen Plateau Ice Caps, Northeastern Ellesmere Island, Nunavut, Canada, follow below. We appreciate the efforts of the reviewer, who clearly spent a great deal of time with this manuscript and as a result have greatly improved it.

Respectfully,

Mark C. Serreze

Director, NSIDC University of Colorado Boulder

[Figure]

Comment:

Overview: This paper documents the important and timely phenomenon of disappearing ice caps in the Canadian high arctic. A time series of area measurements is compiled primarily from previously published observations, with a small contribution of new measurements to document shrinkage and predict the timing of the demise of the plateau ice caps. Some previously published surface mass balance measurements are reported but not discussed. The area changes documented in this study are linked through qualitative comparison to annual temperature 850hPa radiosonde temperatures from Alert. A major shortcoming of this paper is the under-utilization of available long term records from other arctic glaciers to determine if the rate of glacier change over the Hazen plateau is representative or anomalous of the broader scale glacier changes occurring in the Canadian high Arctic. A more rigorous quantitative analysis of the complete time series of changes to the Hazen plateau ice caps should be made in order to contribute to the broader understanding of the rates of climate change in the Canadian high arctic and the current and future rate of contribution of ice caps and glaciers in this region to global sea-level rise.

Response:

We thank the reviewer for his/her efforts. Reviewer 1 and Reviewer 2 have highlighted the same shortcoming of our paper – a failure to adequately place the results from our paper on the context of other studies for the Canadian high Arctic. In response to Reviewer 1 we have made concerted efforts to rectify this problem. This includes comparisons with the efforts by Sharp et al. (2014), Fisher et al. (2012), Wolken et al. (2008) and the mass balance summaries provided in the annual American Meteorological Society State of the Climate Summaries. As noted, providing this fuller context required some reorganization of the text. We feel that the paper is now much more relevant.

Comment:

L58-60: A major shortcoming of this paper is lack of new geophysical data generated for this study, as such, there is no needs for a methods section. The distinction between which data were produced by the authors for this paper, and data/information from previously published material needs to be more clear.

Response:

Correct, we saw no need for a methods section. Apart from the ice cap areas our analysis does not provide new geophysical data, but we do not see this as a weakness. Indeed, by piecing together the old and the new, we have a 55+ year records of the behavior of these ice caps! We have amended the last paragraph to better distinguish what is old and new:

"This paper documents the behavior of the Hazen Plateau ice caps over the past 55+ years in the context of other glaciological studies in the Canadian Arctic. The analysis is based on a combination of past work using aerial photography, direct mass balance measurements from several field investigations, GPS surveys of ice cap areas collected as part of these investigations - along with new information on ice cap areas using data at 15 m resolution from the ASTER (Advanced Spaceborne Thermal Emission and Reflection Radiometer) instrument. ASTER flies onboard the NASA's Earth Observing System Terra satellite, launched in December 1999. It provides reflectance at a 15 m resolution and is a key asset of the international GLIMS initiative (Global Land Ice Measurements from Space) for mapping glacier outlines (Raup et al., 2007; Kargel et al., 2014)".

Comment:

While the authors illustrate the rate of shrinkage of the Hazen plateau ice caps though plots of the time series, there is no attempt made to determine if the rapid changes as determined from this study are occurring at anomalous relative to those documented for other high arctic glaciers.

Response:

Through efforts to place the results from our paper into a broader context, the rapid area reductions of the Hazen Platea ice caps are very much in line with what is happening in the rest of the Arctic. See especially the revisions to Section 3.1.

Comment:

L49: It would be more informative if the actual elevations of the ice cap are rather than just the maximum surrounding area (ie. ": : :ice caps are in an area with maximum elevations s between 750-900 m;: : :") as stated.

Response:

One of the problems here is that the elevation range has changed quite a bit over time. However, we have attempted to add some clarity to the text: "As of 2001, the larger St. Patrick Bay ice cap ranged in elevation between about 880 m and 720 m above sea level, with the smaller one spanning 820 m to 700 m. The Murray and Simmons ice caps lie in higher terrain; in 2001, both fell between about 1100 m and 1000 m above sea level".

Comment:

Extent of LIA glacier cover (Wolken et al.) should be included in the analysis to provide a longer term perspective to the changes discussed in this study.

Response:

See above, this was also pointed out by Reviewer 1 and has been addressed in the first bullet of Section 3.1: "To place these findings in a broader context, for the Queen Elizabeth Islands as a whole, trim lines based on high-resolution satellite imagery point to a 37% reduction in perennial snow and ice extent between the LIA and the year 1960. Over the lower lying central and western islands, a complete removal of perennial snow and ice occurred by 1960 (Wolken et al., 2008)".

Comment:

L53-54: the authors should clarify what form of precipitation this statement (": : :, with a late summer and early autumn maximum..") refers to ie. Rain or snow, or a combination of both, as they can have opposite impacts on the mass balance of small ice caps with no firn to absorb liquid precip.

Response:

Based on personal experience, it can be either. The text has been amended: "Like much of the Queen Elizabeth Islands, the Hazen Plateau is presently a polar desert; annual precipitation is typically only 150-200 mm, with a late summer and early autumn maximum (Serreze and Barry, 2015). Summer precipitation may be variously rain or snow."

Comment:

L83: Presumably these ice caps are stagnant. However, the authors refer to the ice caps ": : :extending its margins,.." which may be misinterpreted as advancing via flow, which is almost certainly not the case. It is most likely that the" extended" margins are actually perennial snow packs which would be of lower density material than the original ice cap. This should be clarified

Response:

This comes directly from the Hattersely-Smith and Serson (1973) paper published in Journal of Glaciology. We have attempted to clarify as follows: "They concluded that while the ice cap had been in decline (as suggested from the 1947 and 1959 photographs), by the early 1970s it had returned to good health, "thickening slightly and extending its margins" (icy firn was observed atop the dirty melt surface and a perennial snow cover extended beyond the ice cap margins).

Comment:

L83: ": : :thickening slightly : : :" how was 'thickening' determined?

Response:

See above, they observed firn atop the former melt surface.

Comment:

L91: "Assuming that the 1982 melt season had largely ended by early August: : :" unless there is temp data to support this claim, there is no reason to assume that the melt season ended in early august. High arctic glaciers at these elevations commonly experience melting into late august.

Response:

Reviewer 1 also pointed this out and the text has been amended accordingly. "Assuming that the 1982 melt season had largely ended by early August (all visible melt had stopped by the time that the field camp had been evacuated), the 1981/1982 mass balance for the larger ice cap was estimated at -0.14 m w.e.. Given that more melt may have occurred, this is likely a minimum estimate." The new Table 1 (the measured mass balances) also indicates that this is a minimum estimate.

Comment:

L94 and 103: it is more informative (and more common in glaciology) to report mass balance as an annual (ie a-1) value even when measurements span multiple years.

Response:

Conversions for periods spanning multiple years have been provided in both the text and in the new Table 1 requested by Reviewer 1.

Comment:

192 " While arguably it might be better to look at the 925 hPa level," the authors need to explain why this is the case.

Response:

Simply put, it is closer to the plateau surface. The text has been amended to point this out.

Comment:

L178/179: the studies referenced refer to loss of ice mass or surface mass balance, not specifically area change. This is an important distinction (and should be discussed) because area reductions of the larger dynamic ice caps are also a function of dynamic response time whereas the margins of small plateau ice caps respond immediately to surface ablation and would shrink at faster rates relative to the dynamic ice masses.

Response:

Reviewer 1 pointed this out as well. We now include more references, which include studies of both mass and area changes. The sentence has been amended: "This is in turn consistent with the broader pattern of reductions in mass and area of Arctic glaciers….."

Comment:

Figure 5: it would be helpful to integrate the annual and multi year average surface mass balance measurements and/or area change values from all studies (this one and all referenced herein) into fig 5 in order to show the relationship between temp change and ice cap response.

Response:

Reviewer 1 wanted to see all of the directly measured mass balances, and in response, we added a new table new (Table 1). Trying to integrate all of the information into Figure 5 proved awkward and crowded. Hence we have compromised, and have indicated on Figure 5 the annual mass balance estimates for the larger St. Patrick Bay (SBP) ice cap for the 1981/1982 and 1982/1983 balance years, and for the Murray Ice Cap (M) for the

1999/2000, 2000/2001, 2001/2002 and 2002/2003 balance years (in m w.e.). We then added some text to more completely discuss the relationships between temperature anomalies and mass balances. The figure caption has also been edited.

Comment:

L54: the Serreze and Barry 2015 is listed as 2014 in the refs.

Response:

It should be 2014; the text has been amended.

Comment:

Figure 2. scale and north arrow unreadable – too small.

Response:

We should have caught this. The scale and north arrow have been made much bigger. We of course edited Figure 3 as well.

Comment:

Figure 1. need to indicate location of Environment Canada weather stations from which data is used. Alert is identified, but should be stated in the caption that it is one location of the long term temp data. Eureka (from which precip data is obtained) is not on the map at all.

Response:

The caption has been amended and the figure has been amended to show station Eureka.
* * *

---

## Author Response (AR1)

Editor, *The Cryosphere*:

Responses to the comments and suggestions to the anonymous reviews of manuscript tc-2016-201 *Rapid Wastage of the Hazen Plateau Ice Caps, Northeastern Ellesmere Island, Nunavut, Canada,* follow below.  We appreciate the efforts of the reviewers, who clearly spent a great deal of time with this manuscript and as a result have greatly improved it.

Respectfully,

Mark  C.  Serreze

Director, NSIDC
University of Colorado Boulder

**Reviewer #1**

This paper documents the near demise of 4 small ice caps in NE Ellesmere Island - ice caps that have a >50 year history of study. The paper is of some interest because of this long history, but it could be made more interesting if more effort were made to place the results in a broader regional context.

We thank the reviewer for his/her obviously very careful read of this paper.  We agree that the paper needed to better place the results into a broader context.  We have attempted to appropriately respond to the reviewer's comments and recommendations that follow below.

Specifically, there is no mention at all of the work of Gabriel Wolken, who used trimline mapping to document the pattern of post Little Ice Age glacier retreat across the Queen Elizabeth Islands and interpret observed patterns in terms of past climate dynamics (The Holocene 18 (4), 615-628 and 629-641, 2008). Nor is there mention of the chapter on this region in the GLIMS book (edited by Jeff Kargel and others; Global Land Ice Measurements from Space, pp 205-228 (Springer, 2014)), which provides a sub-regional breakdown of post 1950's ice retreat across the area as a function of initial ice cap/glacier size. Many small ice caps have disappeared from this region in the past 60 years, but this would not be obvious from reading the submitted manuscript).

Equally, there is no reference to the 4 in situ mass balance time series from the area which provide a nice picture of the short term variability in climate-mass balance linkages that would help with the interpretation of the results presented here - or of the regional ice mass change time series from GRACE, which would do the same thing. The relevant data are published annually in the Arctic section of the

BAMS "State of the Climate" report and are readily available (as are the mass balance time series from WGMS). Nor is any comparison with ice core records from the region, which would also help to provide longer term context for the work (e.g. Fisher et al., 2012, Global and Planetary Change 84, 3-7).

As a result, the paper seems somewhat disconnected from what is already known about post LIA and recent glacier change in the region and its drivers. I think this issue has to be addressed if the paper is to pass the "significance" test for publication in The Cryosphere. The paper does nicely document the history of these specific ice caps in more detail than would be possible for most others in the same region (hence good for originality) , and the detail is sufficient to allow reasonably sophisticated comparison of ice cap retreat rates and patterns with other climate and mass balance indicators for the region - but this is not really attempted (hence fair for scientific quality and significance). This leaves the paper with a rather anecdotal feel. I think this needs to be changed before I could recommend publication in this journal.

We have attempted to place the paper into the broader context of existing research. Note that Reviewer 2 pointed out the same shortcoming.  First, to help set the stage, the first sentence of the second paragraph of the introduction how highlights out intent to place the results into better context: "This paper documents the behavior of the Hazen Plateau ice caps over the past 55+ years in the context of other glaciological studies in the Canadian Arctic."

Next, and importantly, some reorganization was necessary. Specifically, the summarized history of change that was a the end of Section 2 (Previous work) was moved to the end of Section 3.1, and expanded to include a fuller discussion of the history of the ice caps from the LIA to the present (including the area estimates from ASTER through 2016) within the broader context of the studies pointed out above by the reviewer. Section 3 was renamed (Updated History, 1959-2016).

The first bullet of the discussion now includes a comparison with the study of Wolken et al. (2008):

"To place these findings in a broader context, for the Queen Elizabeth Islands as a whole, trim lines based on high-resolution satellite imagery point to a 37% reduction in perennial snow and ice extent between the LIA maximum extent and the year 1960. Over the lower lying central and western islands, a complete removal of perennial snow and ice occurred by 1960 (Wolken et al., 2008)".

The second bullet highlights that the period of reduced mass loss and occasional mass gains from the1960s through at least part of the 1970s is seen across the Canadian Arctic:

"From the 1960s through part of the 1970s, the ice caps may have experienced a period of reduced loss or occasional growth (1971/1972, 1982/1983) in response to cooling.  This basic pattern likely holds for Canadian Arctic glaciers and ice caps as a whole (Bradley and Miller, 1972; Hattersley-Smith and Serson, 1973;  Ommanney, 1977; Bradley and England, 1978; Braun et al., 2004; Sharp et al., 2014)".

The third bullet introduces the persistent subsequent mass losses:

"Since then, apart from occasional years such as 1982/1983, annual mass balances of the four ice caps have been persistently negative (Braun et al., 2004). This is in turn consistent with the broader pattern of reductions in mass and area of Arctic glaciers and ice caps (Dowdeswell et al., 1997; Dyurgerov and Meier, 1997; Arendt et al., 2002; Koerner, 2005; Sharp et al., 2011, 2014; Fisher et al., 2012; Sharp et al., 2014; Mortimer et al., 2016). It is also consistent with a negative mass balance of the Greenland Ice Sheet since at least the 1990s (Shepard et al., 2012)".

This set the stage for the next two (entirely new) paragraphs which discuss specific results from other studies:

"Mass balance summaries for four monitored glaciers and ice caps in the Canadian Arctic (Devon Ice Cap, Meighan Ice Cap, Melville South Ice Cap and the White glacier) are provided as part of the American Meteorological Society State of the Climate reports. As assessed over the period 1980 through 2010, all four have had negative average annual mass balances, ranging from -0.15 m w.e. for the Devon Ice Cap to -0.29 w.e. for the Melville South Ice Cap (AMS, 2016). Cumulative changes in regional total stored water for the period 2003 through 2015 based on gravimetric data from the GRACE mission (Gravity Recovery and Climate Experiment) are qualitatively consistent with these mass balance measurements (AMS, 2016). Based on ice core data, Fisher et al. (2012) document rapid acceleration of ice cap melt rates of over the last few decades across the entire Canadian Arctic; the large reductions in area of the Hazen Plateau ice caps, in particular the lower-elevation St. Patrick Bay ice caps, is consistent with this finding. However, reflecting variable climate conditions, annual balances are also quite variable. For example, for the 2013/2014 balance year (the most recent data available), the White Glacier had a strongly negative balance (-0.42 m w.e.) while the small Meighan Ice Cap actually gained mass (+0.06) (AMS, 2016). Sharp et al. (2014) show that while the larger ice bodies in the Canadian Arctic have seen the larges losses in mass, the smaller masses have lost a larger proportion on their areas. This is also consistent with the behavior of the Hazen Plateau ice caps. Below we examine variability in climate conditions over the Hazen Plateau and links to mass balance and area changes".

Later on, in Section 3.2, Associated Climate Conditions, when speaking of the temperature time series, we note that: "The time series of decadal mean summer temperatures at the 700 hPa level for the major glaciated regions of the Canadian Arctic presented by Sharp et al. (2011) based on the NCEP/NCAR reanalysis (their Figure 9.3) is broadly consistent with the pattern shown in Figure 5".

The State of the Climate published by the American Meteorological Society that the reviewer recommended also provide some useful insight into the 2012/2103 mass balance year, now discussed in the last paragraph of Section 3.2, in which we call a new figure (Figure 6) showing the summer 825 hPa temperature anomaly pattern:

"Regarding the summer of 2013, the obvious exception to the pattern of recent warm years, the ASTER data and daily images from the Moderate Resolution Imaging Spectroradiometer (MODIS) show extensive cloud cover through the summer, making it difficult to determine whether the snow cover ever entirely cleared off the plateau. It is likely, however, that the 2012/2013 balance year was positive for the Hazen Plateau ice caps - the Devon Ice Cap, Meighan Ice Cap and the White Glacier all gained mass. Only the Melville South Ice Cap, lying well to the west, had a negative balance (AMS, 2014). Consistent with this view, **Figure 6** shows that summer (J,J,A) averaged 850 hPa temperature anomalies over the Queen Elizabeth Islands from the NCEP/NCAR reanalysis were about 2$^o$C below the 1981-2010 baseline in the area centered over Axel Heiberg and Ellesmere islands. This reflects the influence of an unusually deep circumpolar vortex at the 500 hPa level centered just south of the Pole along about 90$^o$W longitude. By sharp contrast, the notable area reduction of the St. Patrick Bay ice caps between August 2014 and 2015 aligns with the very warm summer of 2015, essentially tied with 1957 as the highest in the record. From **Figure 7**, July 2015 temperatures at the 850 hPa level from the NCEP/NCAR reanalysis were 3-4$^o$C above the 1981-2010 baseline over most of northeastern Ellesmere Island. Mass balance estimates for monitored glaciers in the Queen Elizabeth Islands for the 2014/2015 season that would provide context were not available us at the time that this paper came to press".

Note that Figure 6 and 7 are presented at a scale that shows conditions over all of the Queen Elizabeth Islands instead of focusing (as the original Figure 6 did) just on Ellesmere Island.

The paper is however quite readable (good for presentation quality) although I do suggest below quite a number of detailed edits that would make it more readable.

We thank the reviewer for these suggested edits and additions.

Specific Points (keyed by line number):

71-74: Specify uncertainties associated with the ice cap area estimates - important to know how large these are relative to the observed changes

Text has been added: "We estimate that these area estimates are accurate to within 5%".

77: what is the range of surface elevations covered by this transect and how does it compare with the total elevation range of the ice cap?

Additional text has been added to the paragraph: "The range in elevation along this transect was about 60 m, which compares to a range for the entire ice cap of about 160 m."

85-86: this statement seems like unnecessary speculation, given that the comparison is with the behaviour of a single studied ice cap.

The sentence has been cut.

91: Why assume the 1982 melt season ended by the end of August ? Evidence for this claim? Climate data?

It is of course possible that more melt occurred, although all visible melt had stopped by the time that the field camp had been evacuated and the daily maximum air temperature drops rapidly in late August. However, given that more melt may have occurred, the estimate mass balance of -0.14 w.e. is likely a minimum estimate. The text has been revised to note this. Reviewer 2 also pointed this out.

92-94: an annual mean MB value for a given time interval might be more useful than a period mean and, if there is a stake line, it would be useful to say something about how the annual balance varies with elevation.

The elevation range of the ice cap is quite small and as such, based on the first author's field notes, no variation in the mass balance with elevation was apparent.

96: decreased in area (rather than shrunk in area)

Corrected.

103: Is -0.49 the annual value or a period mean? Not clear.

It is the total over the period. The text has been amended to: Based on these sparse data, Bradley and Serreze (1987) estimated that over the period 1976-1983, the Simmons ice cap experienced a total mass loss of at least -0.49 meters water equivalent.

104: inferred by Hattersley-Smith.....

Corrected.

106-107: Treats temperature and mass balance as interchangeable terms - not justified, so talk about temperature as that is what the data relate to (or present a quantitative relationship that justifies inferring MB from T

This was poor wording on our part. The paragraph now reads: "However, the summer of 1983 was fairly cool and the snow never completely melted off the surrounding tundra. The 1982/1983 annual mass balance for the larger St. Patrick Bay ice cap was estimated at +0.14 m water equivalent, and given their higher elevation, it is reasonable to assume that the 1982/1983 balance of the Simmons and Murray ice caps was also positive".

109: 1982/83 balance of the Simmons.....

Corrected.

116-117: annual balances of both ice caps were negative in all years, ranging from....

Corrected.

120: The larger and smaller ..of their 1959 areas respectively, while the

Corrected.

Murray...had shrunk to 70%......their 1957 areas

Corrected.

123: inserted allowed us to make a minimum estimate (-1.10 m w.e.) of the mass loss...

Corrected.

125-128: given the measurement approach, I think some assessment of the associated errors is warranted.

Text has been added to the paragraph: "This is based on the mean remaining depth of stake insertion into the ice in 1983 and an assumed ice density of 900 kg m (Braun et al., 2004)".

129:……studies, and the results from…..

Corrected.

131: maximum, and likely to have formed

It seems better as "…..but rather likely formed…"

133: in the first couple….

Corrected.

134-135: On the basis of a mapped lichen tramline, Braun et al. (2004) speculate that

Corrected.

135-138: Here the authors should make reference to Wolken's study (The Holocene) of lichen trimlines in the QEI; I'd also like to see a tabulation of all the available surface mass balance measurements and the time periods they represent.

The section has been modified as follows: "Braun et al. (2004) speculate on the basis of a mapped lichen trim line that the Murray ice cap may have attained a maximum LIA extent of about 9.6 km$^2$, over twice the mapped 1959 area of 4.35 km$^2$. Similar trim lines were observed around the other three ice caps and although not mapped in detail, strongly point to much more extensive ice cover during the LIA. To place these findings in a broader context, for the Queen Elizabeth Islands as a whole, trim lines based on high-resolution satellite imagery point to a 37% reduction in perennial snow and ice extent between the LIA maximum extent and the year 1960. Over the lower-lying central and western islands, a complete removal of perennial snow and ice occurred by 1960 (Wolken et al., 2008)".

A new table (Table 1) has been added that summarizes all available direct mass balance estimates of the ice cap. Note that the similar table of Braun et al. (2004) includes some additional estimates for other year based on indirect approached.

143: List all the known positive balance years so we can see how many there have been

Done.

147: Note there are GRACE mass balance time series for the QEI (see the Arctic section of successive annual BAMS "State of the Climate" reports) and for the Russian Arctic, Svalbard, and Alaska, so you could make broader scale Arctic comparisons with your data time series.

We now include mention of the GRACE results in the revised Section 3.1.

152: areas (in km2) from all…

Corrected.

157: of the Murray

Corrected.

162: 2016, the Murray…….in 1959. By sharp contrast, ….…..only 0.15 km2

Corrected.

168: reductions in…..area are striking

Corrected.

170: is shown in Figure 3, based on outlines from 1959, 2001, and 2016.

Corrected.

177: Note that none of these studies discuss glacier area changes, and you don't reference the one that does (paper in the GLIMS book)

We now include more references, which include studies of both mass and area changes.   The sentence has been amended:  "This is in turn consistent with the broader pattern of reductions in mass and area of Arctic glaciers….."

197: is it meaningful to make comparisons with pan-Arctic means given the scale of this study?

We clearly state that we are making an assumption that the inferred LIA conditions over the Hazen Plateau from the study of Kaufman et al. (2011), are at least broadly similar to those for the Arctic as a whole. It is not clear what more can be done here to obtain an optimal local estimate.

200-203: Sharp et al (2011) provide evidence that would support this assumption (i.e. that 850 hPa and surface air temperatures show similar patterns)

The text has amended to point this out.

**Reviewer #2**

Overview: This paper documents the important and timely phenomenon of disappearing ice caps in the Canadian high arctic. A time series of area measurements is compiled primarily from previously published observations, with a small contribution of new measurements to document shrinkage and predict the timing of the demise of the plateau ice caps. Some previously published surface mass balance measurements are reported but not discussed. The area changes documented in this study are linked through qualitative comparison to annual temperature 850hPa radiosonde temperatures from Alert. A major shortcoming of this paper is the under-utilization of available long term records from other arctic glaciers to determine if the rate of glacier change over the Hazen plateau is representative or anomalous of the broader scale glacier changes occurring in the Canadian high Arctic. A more rigorous quantitative analysis of the complete time series of changes to the Hazen plateau ice caps should be made in order to contribute to the broader understanding of the rates of climate change in the Canadian high arctic and the current and future rate of contribution of ice caps and glaciers in this region to global sea-level rise.

We thank the reviewer for his/her efforts.   Reviewer 1 and Reviewer 2 have highlighted the same shortcoming of our paper  – a failure to adequately place the results from our paper on the context of other studies for the Canadian high Arctic.  In response to Reviewer 1 we have made concerted efforts to rectify this problem.   This includes comparisons with the efforts by Sharp et al. (2014), Fisher et al. (2012), Wolken et al. (2008) and the mass balance summaries provided in the annual American Meteorological Society State of the Climate Summaries.   As noted, providing this fuller context required some reorganization of the text.  We feel that the paper is now much more relevant.

Comments: L58-60: A major shortcoming of this paper is lack of new geophysical data generated for this study, as such, there is no needs for a methods section. The distinction between which data were produced by the authors for this paper, and data/information from previously published material needs to be more clear.

Correct, we saw no need for a methods section.   Apart from the ice cap areas our analysis does not provide new geophysical data, but we do not see this as a weakness.  Indeed, by piecing together the old and the new, we have a 55+ year records of the behavior of these ice caps!  We have amended the last paragraph to better distinguish what is old and new:

"This paper documents the behavior of the Hazen Plateau ice caps over the past 55+ years in the context of other glaciological studies in the Canadian Arctic.   The analysis is based on a combination of past work using aerial photography, direct mass balance measurements from several field investigations, GPS surveys of ice cap areas collected as part of these investigations - along with new information on ice cap areas using data at 15 m resolution from the ASTER (Advanced Spaceborne Thermal Emission and Reflection Radiometer) instrument.  ASTER flies onboard the NASA's Earth Observing System Terra satellite, launched in December 1999.  It provides reflectance at a 15 m resolution and is a key asset of the international GLIMS initiative (Global Land Ice Measurements from Space) for mapping glacier outlines (Raup et al., 2007; Kargel et al., 2014)".

While the authors illustrate the rate of shrinkage of the Hazen plateau ice caps though plots of the time series, there is no attempt made to determine if the rapid changes as determined from this study are occurring at anomalous relative to those documented for other high arctic glaciers.

Through efforts to place the results from our paper into a broader context, the rapid area reductions of the Hazen Platea ice caps are very much in line with what is happening in the rest of the Arctic.  See especially the revisions to Section 3.1.

L49: It would be more informative if the actual elevations of the ice cap are rather than just the maximum surrounding area (ie. ": : :ice caps are in an area with maximum elevations s between 750-900 m;: : :") as stated.

One of the problems here is that the elevation range has changed quite a bit over time.  However, we have attempted to add some clarity to the text:  "As of 2001, the larger St. Patrick Bay ice cap ranged in elevation between about 880 m and 720 m above sea level, with the smaller one spanning 820 m to 700 m.  The Murray and Simmons ice caps lie in higher terrain; in 2001, both fell between about 1100 m and 1000 m above sea level".

Extent of LIA glacier cover (Wolken et al.) should be included in the analysis to provide a longer term perspective to the changes discussed in this study.

See above, this was also pointed out by Reviewer 1 and has been addressed in the first bullet of Section 3.1:  "To place these findings in a broader context, for the Queen Elizabeth Islands as a whole, trim lines based on high-resolution satellite imagery point to a 37% reduction in perennial snow and ice extent between the LIA and the year 1960. Over the lower lying central and western islands, a complete removal of perennial snow and ice occurred by 1960 (Wolken et al., 2008)".

L53-54: the authors should clarify what form of precipitation this statement (": : :, with a late summer and early autumn maximum..") refers to ie.  Rain or snow, or a combination of both, as they can have opposite impacts on the mass balance of small ice caps with no firn to absorb liquid precip.

Based on personal experience, it can be either.  The text has been amended:  "Like much of the Queen Elizabeth Islands, the Hazen Plateau  is presently a  polar desert;  annual precipitation is typically only 150-200 mm, with a late summer and early autumn maximum (Serreze and Barry, 2015).  Summer precipitation may be variously rain or snow."

L83: Presumably these ice caps are stagnant. However, the authors refer to the ice caps ": : :extending its margins,.." which may be misinterpreted as advancing via flow, which is almost certainly not the case. It is most likely that the" extended" margins are actually perennial snow packs which would be of lower density material than the original ice cap. This should be clarified

This comes directly from the Hattersely-Smith and Serson (1973) paper published in *Journal of Glaciology*.  We have attempted to clarify as follows:  "They concluded that while the ice cap had been in decline (as suggested from the 1947 and 1959 photographs), by the early 1970s it had returned to good health, "thickening slightly and extending its margins" (icy firn was observed atop the dirty melt surface and a perennial snow cover extended beyond the ice cap margins).

L83: ": : :thickening slightly : : :" how was 'thickening' determined?

See above, they observed firn atop the former melt surface.

L91: "Assuming that the 1982 melt season had largely ended by early August: : :" unless there is temp data to support this claim, there is no reason to assume that the melt season ended in early august. High arctic glaciers at these elevations commonly experience melting into late august.

Reviewer 1 also pointed this out and the text has been amended accordingly. "Assuming that the 1982 melt season had largely ended by early August (all visible melt had stopped by the time that the field camp had been evacuated), the 1981/1982 mass balance for the larger ice cap was estimated at -0.14 m w.e.. Given that more melt may have occurred, this is likely a minimum estimate." The new Table 1 (the measured mass balances) also indicates that this is a minimum estimate.

L94 and 103: it is more informative (and more common in glaciology) to report mass balance as an annual (ie a-1) value even when measurements span multiple years.

Conversions for periods spanning multiple years have been provided in both the text and in the new Table 1 requested by Reviewer 1.

" While arguably it might be better to look at the 925 hPa level," the authors need to explain why this is the case.

Simply put, it is closer to the plateau surface. The text has been amended to point this out.

L178/179: the studies referenced refer to loss of ice mass or surface mass balance, not specifically area change. This is an important distinction (and should be discussed) because area reductions of the larger dynamic ice caps are also a function of dynamic response time whereas the margins of small plateau ice caps respond immediately to surface ablation and would shrink at faster rates relative to the dynamic ice masses.

Reviewer 1 pointed this out as well. We now include more references, which include studies of both mass and area changes. The sentence has been amended: "This is in turn consistent with the broader pattern of reductions in mass and area of Arctic glaciers….."

Figure 5: it would be helpful to integrate the annual and multi year average surface mass balance measurements and/or area change values from all studies (this one and all referenced herein) into fig 5 in order to show the relationship between temp change and ice cap response.

Reviewer 1 wanted to see all of the directly measured mass balances, and in response, we added a new table new (Table 1). Trying to integrate all of the information into Figure 5 proved awkward and crowded. Hence we have compromised, and have indicated on Figure 5 the annual mass balance estimates for the larger St. Patrick Bay (SBP) ice cap for the 1981/1982 and 1982/1983 balance years, and for the Murray Ice Cap (M) for the 1999/2000, 2000/2001, 2001/2002 and 2002/2003 balance years (in m w.e.). We then added some text to more completely discuss the relationships between temperature anomalies and mass balances. The figure caption has also been edited.

L54: the Serreze and Barry 2015 is listed as 2014 in the refs.

It should be 2014; the text has been amended.

Figure 2. scale and north arrow unreadable – too small.

We should have caught this. The scale and north arrow have been made much bigger. We of course edited Figure 3 as well.

Figure 1. need to indicate location of Environment Canada weather stations from which data is used. Alert is identified, but should be stated in the caption that it is one location of the long term temp data. Eureka (from which precip data is obtained) is not on the map at all.

The caption has been amended and the figure has been amended to show station Eureka.

[revised manuscript text omitted]

---

## Author Response (AR2)

Editor, *The Cryosphere*:

January 3, 2017

Responses to the remaining comments and suggestions to the anonymous reviews of manuscript tc-2016-201 *Rapid Wastage of the Hazen Plateau Ice Caps, Northeastern Ellesmere Island, Nunavut, Canada,* follow below.  Again, we appreciate the efforts of the reviewers, who clearly spent a great deal of time with this manuscript and as a result have greatly improved it.

**Reviewer #1**: The reviewer caught 3 remaining typos in the manuscript, these have been fixed.
**Reviewer #2:**  The reviewer noted that Figure 5 still needed work, including the addition of proper labels, completion of the plot area box and overlapping text.  These issues have been addressed. Figure 5 is much better now.

Respectfully,

Mark  C.  Serreze

Director, NSIDC
University of Colorado Boulder

[revised manuscript text omitted]